# Payments and freedoms: Effects of monetary and legal incentives on COVID-19 vaccination intentions in Germany

**Philipp Sprengholz**[1,2]*, **Luca Henkel**[3], **Cornelia Betsch**[1,3,4]

**1** Media and Communication Science, University of Erfurt, Erfurt, Germany, **2** Health Communication, Bernhard Nocht Institute for Tropical Medicine, Hamburg, Germany, **3** Center for Empirical Research in Economics and Behavioral Sciences, University of Erfurt, Erfurt, Germany, **4** University of Bonn, Bonn, Germany

* philipp.sprengholz@uni-erfurt.de

**Data Availability Statement:** Supplemental tables, data, and the data analysis script are available at https://dx.doi.org/10.17605/OSF.IO/4KW2U.

## Abstract

Monetary and legal incentives have been proposed to promote COVID-19 vaccination uptake. To evaluate the suitability of incentives, an experiment with German participants examined the effects of payments (varied within subjects: 0 to 10,000 EUR) and freedoms (varied between subjects: vaccination leading vs. not leading to the same benefits as a negative test result) on the vaccination intentions of previously unvaccinated individuals ($n =$ 782) in April 2021. While no effect could be found for freedoms, the share of participants willing to be vaccinated increased with the payment amount. However, a significant change required large rewards of 3,250 EUR or more. While monetary incentives could increase vaccination uptake by a few percentage points, the high costs of implementation challenge the efficiency of the measure and call for alternatives. As the data suggest that considering vaccination as safe, necessary, and prosocial increases an individual's likelihood of wanting to get vaccinated without payment, interventions should focus on these features when promoting vaccination against COVID-19.

## Introduction

Rapid, large-scale uptake of vaccines against COVID-19 is required to control and eventually end the SARS-CoV-2 pandemic. However, vaccine hesitancy may prevent a significant share of the population from getting vaccinated [1]. For example, as of March 2022, only 76% of the German population and 77% of Americans had received at least one shot against COVID-19, with very low increases in uptake since autumn 2021 [2]. Previous research indicates that low vaccination intentions can result from people having little confidence in vaccine safety, being complacent (i.e., considering vaccination as rather unnecessary) or calculative (i.e., extensively weighing risks and benefits), encountering barriers constraining the act of getting vaccinated, or perceiving low collective responsibility (e.g., lack of willingness to get vaccinated to protect others) [3]. Researchers have discussed various interventions for addressing these antecedents of vaccination and increasing vaccination intentions, ranging from information campaigns to

**Funding:** This work was supported by German Research Foundation (BE3970/12-1), Federal Centre for Health Education, Robert Koch Institute, Leibniz Institute for Psychology, Klaus Tschira Foundation, and University of Erfurt (no award/ grant numbers). The funders had no role in study design, data collection and analysis, decision to publish, or preparation of the manuscript.

**Competing interests:** The authors have declared that no competing interests exist.

mandatory vaccination [4–6]. As incentives have been shown to promote certain health behaviors, such as maintaining a healthier diet and quitting smoking [7, 8], offering rewards for vaccination could bolster vaccination intentions as well. This may be done in different ways.

First, monetary incentives could be used, reimbursing people for the time needed to get vaccinated and to recover from possible side effects. Systematic reviews differ in their recommendation of monetary incentives. While the Community Preventive Services Task Force [9] recommended them based on studies about influenza, tetanus, diphtheria and pertussis as well as childhood vaccination, Adams et al. [10] could not find sufficient evidence to recommend parental financial incentives for vaccination of preschool children. The inconclusive evidence may reflect the heterogeneity of studies [11] and requires further investigation for the new vaccines against COVID-19. Despite scarce knowledge about the effects of vaccination incentives, multiple companies in the United States and Germany started offering employees one-time payments for vaccination against COVID-19 once vaccines became available [12], and in July 2021, the White House called on state governments to pay 100 USD to those who are willing to get vaccinated [13]. While a hypothetical experiment with German participants at the end of 2020 suggested that payments of up to 200 EUR (about 240 USD) for getting vaccinated did not increase people's intentions to get vaccinated [14], a different picture emerged in another German study conducted in March 2021, where hypothetical vouchers worth 25 EUR or 50 EUR (about 60 USD) increased the subjective probability of getting vaccinated by 1 or 2.2 percentage points [15]. Evidence from the US was inconclusive as well. While one study showed that vaccination intentions increased by 4.5 (13.6) percentage points after individuals were offered a hypothetical payment of 100 (500) USD [16], another article investigating the effects of real payments of 10 and 50 USD in May and July 2021 found insignificant overall effects [17]. Data from Sweden further showed that offering a 24 USD voucher increased vaccination uptake by as much as 4 percentage points [18]. While the effect of monetary incentives certainly depends on the pandemic context, the current uptake level, the local health system and cultural background, the heterogeneity of results may in part be explained by payment size. Offering larger incentives could be more effective than paying smaller sums for increasing vaccination intentions. Accordingly, some researchers advocated for giving people 1,000 USD [19] and companies like Coca Cola started paying employees 2,000 USD for getting vaccinated [20].

As a second possibility to increase vaccination uptake, legal incentives could be employed. As vaccinated individuals are less likely to transmit some virus variants [21] and are less likely to suffer from severe infections requiring hospitalization [22, 23], they could enjoy more rights and freedoms and be less constrained by COVID-19 regulations compared to unvaccinated people. For instance, allowing vaccinated but not unvaccinated individuals to enter shops, get haircuts, or attend certain events without having to get tested could drive vaccination intentions [24]. However, there is little research on the effects of freedoms. One study [15] showed that offering hypothetical freedoms for vaccination (such as being allowed to travel, visiting cinemas, restaurants, or concerts) increased the subjective probability of getting vaccinated by 2.5 percentage points in a German study conducted in March 2021. Given this rather small effect, the impact of legal incentives may be negligible, especially when the offered freedoms are less extensive, and when they can also be gained with a negative test (as was the case in Germany in most of 2020 and 2021).

While researchers have discussed the ethics of both monetary and legal incentives for COVID-19 vaccine uptake [25–28] but empirical research on the effects of both interventions was inconclusive or unavailable, we conducted a survey experiment investigating their single and combined potential effects on binary vaccination intentions in April 2021 in Germany. At that time, multiple vaccines were approved and recommended for the general population, but

access was still restricted. As vulnerable groups and healthcare personnel were prioritized for vaccination, large parts of the population had to wait for their first shot, allowing us to investigate the impact of small and large payments as well as freedoms on vaccination intentions. While the pandemic situation evolves rapidly and attitudes towards vaccination and policy interventions may have changed since the experiment, the evidence can still help in weighing the benefits and costs of monetary and legal incentives and, thus, can inform the efficient design of future vaccination policies.

## Methods

### Participants and design

The experiment was conducted on April 20–21, 2021, as part of the COVID-19 Snapshot Monitoring (COSMO) cross-sectional online study series [29]. Participants were recruited from a non-probabilistic German sample ($N$ = 997), which was quota-representative for age × gender and federal state. Excluding participants who had already been vaccinated against COVID-19 ($n$ = 215) yielded a final sample of $n$ = 782. Participants ranged in age from 18 to 74 years ($M$ = 44.01, $SD$ = 15.66) and included 376 males and 406 females. They were randomly assigned to one of two experimental conditions (legal incentive vs. no legal incentive), and financial incentives were manipulated within subjects.

The study obtained ethical clearance from the University of Erfurt's IRB (#20200302/ 20200501), and all participants provided informed consent prior to data collection.

### Procedure and materials

We assessed relevant demographic information, financial worries, and psychological antecedents of vaccination for explorative purposes. After participants had read a scenario based on the legal incentive condition, they repeatedly decided between not getting vaccinated and getting vaccinated with an incentive ranging from 0 to 10,000 EUR.

**Financial worries.**   Participants were asked how worried they are about getting into financial trouble over losing money due to the pandemic. Answers were assessed on a scale ranging from 1 (not worried at all) to 7 (very much worried).

**Psychological antecedents of vaccination.**   Participants were asked to think about a COVID-19 vaccine that was officially recommended for them. An adapted version of the 5C short scale [3] was used to assess confidence (*I am completely confident that the COVID-19 vaccine is safe*), complacency (*Vaccination against COVID-19 is unnecessary because COVID-19 is not common anymore*), constraints (*Everyday stress prevents me from getting vaccinated against COVID-19*), calculation (*When I think about getting vaccinated against COVID-19, I weigh benefits and risks to make the best decision possible*), and collective responsibility (*When everyone is vaccinated against COVID-19, I don't have to get vaccinated too*) in relation to the vaccine. Items were rated on a scale ranging from 1 (strongly disagree) to 7 (strongly agree). Scores for collective responsibility were reversed before analyses.

**Experimental manipulation.**   At the time of the experiment, legal regulations in Germany required everyone to wear face masks in public areas, such as city centers, and it was mandatory to have a negative coronavirus test to attend cultural events (when permitted at all) or to access services, such as haircuts. Participants in the legal incentive condition were asked to imagine that being vaccinated would lead to more rights in those areas, such as being allowed to discontinue wearing face masks and not needing a test to attend cultural events or access services. In the no legal incentive condition, participants were told that getting vaccinated would not result in additional freedoms.

**Vaccination decisions.** A price list design was used to determine participants' vaccination intentions and payment preferences. Price lists are a standard method for measuring the effects of monetary incentives, since they are easy to explain and implement [30] and have been used in many contexts [31, 32]. In a series of repeated decisions, participants chose between two options: not getting vaccinated vs. getting vaccinated and being paid a specific amount. Amounts from 0 EUR to 5,000 EUR (in increments of 250 EUR) and 10,000 EUR were offered, resulting in 22 binary decisions for each participant. We decided against including minor amounts below 250 EUR since previous research indicated that monetary incentives up to 200 EUR do not affect vaccination intentions of the German population [14].

## Results

### Effects of legal and monetary incentives on vaccination intentions

Fig 1 displays the fraction of participants willing to get vaccinated for each monetary amount offered in the price list. Legal incentives had virtually no impact on vaccination intentions. Without payment (0 EUR), 61.4% of participants in the no legal incentive condition and 65.1% in the legal incentive condition were willing to get vaccinated; the difference was not significant ($p$ = .300; two-sided Fisher's exact test). Similarly, differences were not significant for every other monetary amount except for 10,000 EUR. Interestingly, the share of people willing to get vaccinated for this large reward was higher when no legal incentives were offered ($p$ = 0.043; two-sided Fisher's exact test). Notably, however, the latter significant difference was not robust to the exclusion of 84 participants (10.7%) with non-monotone vaccination intentions—that is, those who opted for vaccination at some amount but switched to non-

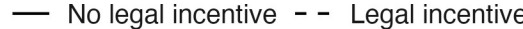

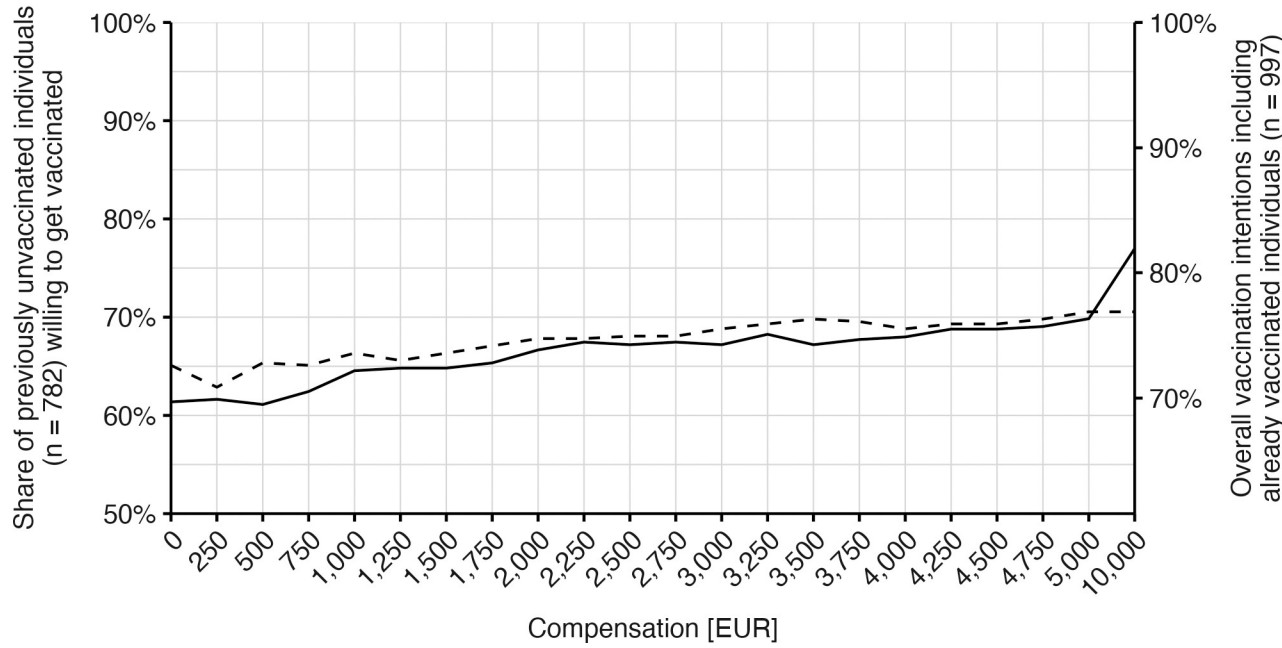

**Fig 1. Willingness to get vaccinated by legal and monetary incentives.** While legal incentives had virtually no impact on vaccination intentions, monetary incentives of 3,250 EUR and above led to a significant increase of people willing to be vaccinated against COVID-19 (compared to 0 EUR). As a reference, the fraction on the right also includes the $n$ = 215 participants that had already been vaccinated and were not included in the experiment.

vaccination at a higher amount (see online supplement). In total 44 participants (6%) changed intentions multiple times, i.e., switched between vaccination and non-vaccination more than once. The share of these multiple switchers did not differ between legal incentive conditions ($p$ = .12; Fisher's exact test).

Monetary incentives increased participants' willingness to get vaccinated, but for significant increases, large amounts were needed. At the 5% level, a significant difference compared to the 0 EUR benchmark was only reached at 3,250 EUR and above. When presented with the maximum offer of 10,000 EUR, the share of people willing to get vaccinated increased by 10.4 percentage points compared to when no money was offered.

The results were in line with a supplemental linear regression investigating the main and interaction effects of payments (with legal incentives, the 5C, and socio-demographic variables) on vaccination intentions, again showing that legal incentives play no important role while payments can have a positive impact (for details, see S1 Table).

## Determinants of getting vaccinated with and without monetary incentives

To investigate factors associated with the decision to get vaccinated with or without monetary incentives, we divided the experimental sample into three groups: (1) participants not willing to get vaccinated regardless of whether payment was offered ($n$ = 144), (2) participants willing to get vaccinated without payment ($n$ = 495), and (3) participants willing to get vaccinated only when payment was offered ($n$ = 143). Participants with non-monotone intentions and those who switched between vaccination and non-vaccination multiple times were included in the latter two groups, i.e., when intending to get vaccinated without payment but denying vaccination for money in (2), or when intending to get vaccinated for some amount (different from 0 EUR) but opting for non-vaccination at a higher amount in (3). A multinomial logistic regression was performed to investigate differences among the three groups regarding age, gender, financial worries, the 5C, and the impact of legal incentives (Table 1), further controlling for household size and income, education, and migration background (for complete results, see S2 Table).

Participants who preferred being vaccinated (regardless of payment preferences) indicated higher levels of confidence and lower levels of complacency compared to those who did not want

**Table 1. Determinants of getting vaccinated with and without monetary incentive.**

| Predictors | Getting vaccinated without monetary incentive | | Getting vaccinated for monetary incentive only | |
|---|---|---|---|---|
| | *OR* | *95% CI* | *OR* | *95% CI* |
| (Constant) | **0.06** | 0.006–0.648 | 1.13 | 0.146–8.715 |
| Experimental manipulation: legal incentive (Baseline: no legal incentive) | 1.01 | 0.577–1.755 | 0.62 | 0.362–1.059 |
| Age | 1.01 | 0.984–1.027 | **0.98** | 0.959–0.999 |
| Gender: female (Baseline: male) | 0.63 | 0.347–1.129 | 0.79 | 0.442–1.402 |
| Financial worries | 1.02 | 0.896–1.156 | 1.02 | 0.913–1.156 |
| Confidence | **2.32** | 1.942–2.771 | **1.49** | 1.257–1.769 |
| Complacency | **0.55** | 0.451–0.683 | **0.81** | 0.688–0.957 |
| Calculation | **0.80** | 0.677–0.935 | 0.99 | 0.851–1.159 |
| Constraints | 1.09 | 0.879–1.358 | 1.06 | 0.894–1.289 |
| Collective responsibility | **1.51** | 1.259–1.804 | 0.99 | 0.851–1.151 |

Results of the multinomial logistic regression analysis (Cox & Snell's $R^2$ = .51, Nagelkerke's $R^2$ = .61). Both groups were compared to participants not willing to get vaccinated regardless of payment. Results were further controlled for household size and income, education, and migration background (for complete results, see S2 Table). Bold values denote significant predictors with $p$ < .05.

to get vaccinated. When comparing the two groups willing to get vaccinated, those who preferred payment showed less confidence and more complacency. Furthermore, those who were willing to get vaccinated without payment indicated less calculation and higher levels of collective responsibility compared to the other two groups. While there were no differences between the groups regarding gender and financial worries, younger participants preferred a financial incentive for vaccination. As before, legal incentives did not predict participants' vaccination decisions.

## Determinants of minimum required monetary incentives

We further examined factors influencing the minimum required monetary incentive for participants who were willing to get vaccinated only if payment was offered ($n$ = 143). Two linear regressions were conducted to regress the minimum accepted payment on age, gender, financial worries, the 5C, and legal incentives, again controlling for household size and income, education, and migration background. The two regressions dealt differently with participants who switched between non-vaccination and (paid) vaccination more than once ($n$ = 28). In the first regression (model 1), these participants were excluded; in the second regression (model 2), their first switching point was interpreted as their minimum accepted payment and served as a dependent variable. In both regressions, no significant effects could be found for gender, financial worries, and the 5C (see S3 Table). However, in both regressions, larger payments were related to higher age (model 1: $\beta$ = 0.24, $b$ = 60.08, $SE$ = 26.81, $95\%$ $CI$ = [7.531; 112.619]; model 2: $\beta$ = 0.26, $b$ = 60.76, $SE$ = 22.39, $95\%$ $CI$ = [16.885; 104.637]) and not being offered legal incentives (model 1: $\beta$ = −0.35, $b$ = −2,620.53, $SE$ = 744.76, $95\%$ $CI$ = [−4,080.235; −1,160.826]; model 2: $\beta$ = −0.37, $b$ = −2,550.45, $SE$ = 561.49, $95\%$ $CI$ = [−3,650.939; −1,449.957]). Furthermore, in model 2, stronger perceived constraints to getting vaccinated were related to higher monetary incentives ($\beta$ = 0.20, $b$ = 395.50, $SE$ = 198.72, $95\%$ $CI$ = [6.007; 784.992]). It should be noted that model 1 was slightly underpowered to find small to medium effects. When considering an error probability of $\alpha$ = .05, a power of $1 - \beta$ = .90, and 9 predictors, at least 141 participants were required to find medium effects with $f^2$ = .15. While this limits the interpretability of insignificant findings in model 1, model 2 was well-powered and the reported findings should be considered robust.

## Extrapolated costs of monetary incentives

As monetary incentives had a positive effect on vaccination intentions, we calculated implementation costs when offering the adult German population a one-time payment for initial vaccination (18 years and older, including those who have already been vaccinated; about 70 million people). Based on the aforementioned results, costs depended on the targeted vaccination uptake rate (Fig 2). While it would be possible to vaccinate about 70% of the adult population without payments, increasing vaccination uptake to 80%, as demanded by the World Health Organization at the time of our study (WHO, 2021), would require incentives worth at least 500 billion EUR.

## Discussion

We investigated the effects of legal and monetary incentives on vaccination intentions. Our results indicate that legal incentives do not increase the willingness to vaccinate. However, this may be due to the specific incentives offered in the experiment, where vaccination was predominantly framed as a replacement for testing. Offering stronger incentives, such as being allowed to eat out, travel for leisure, or attend a music festival, could indeed boost vaccination intentions as indicated by other experimental research [15] and a recent analysis showing that

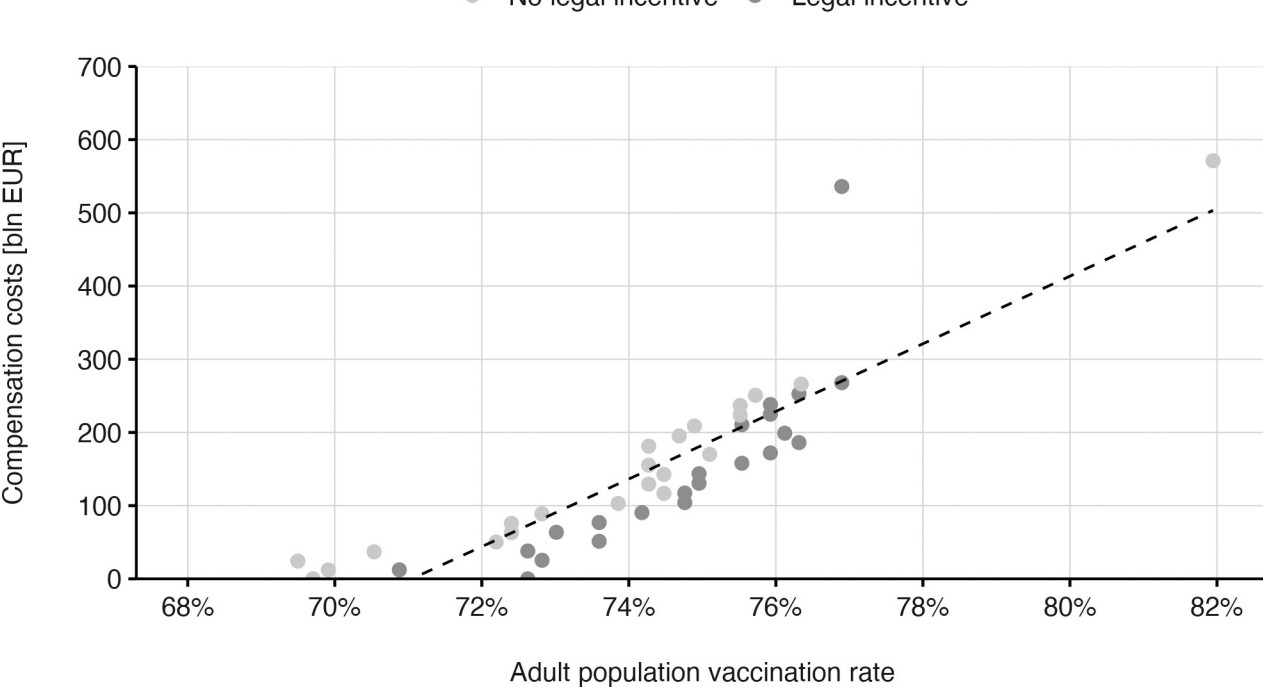

**Fig 2. Economic costs associated with realizing specific vaccination uptake rates for the adult population in Germany.** The visualization assumes that individuals 75 years and older will make decisions similar to the younger adults examined in our study. The dotted line denotes a linear fit after collapsing the legal incentive conditions.

the introduction of vaccination certificates required to access specific venues such as restaurants and clubs in late 2021 increased vaccination rates in Italy, France, and Germany [33].

While the majority of previously unvaccinated participants were willing to get vaccinated without a financial reward, about a fifth opted for vaccination only when a payment of up to 10,000 EUR was offered. Interestingly, compared to participants who were willing to get vaccinated regardless of payment, monetary incentives motivated less confident and more complacent participants to want the vaccine. Thus, people who think that vaccination is rather unnecessary and who are not entirely sure about the safety of vaccines could be motivated to get vaccinated when (high) monetary incentives are offered.

The WHO [34] has urged countries to vaccinate at least 80% of their adult population as soon as feasible. Our results indicate that monetary incentives could help to achieve this rate in Germany. However, more people may need to be vaccinated when more contagious mutations of the virus emerge. In addition, immunizing children will become important for the same reasons. As previous research indicates that parents are more risk averse when contemplating their children's vaccination than their own [35], the impact of monetary incentives on these decisions may be small. Furthermore, paying parents to get their children vaccinated may be ethically questionable, calling for other measures to improve vaccine uptake.

Overall, the data revealed that high amounts need to be paid to make a difference in Germany. While Serra-Garcia and Szech [16] showed that vaccination uptake among Americans could be leveraged by more than 10% by offering a payment of 500 USD at the beginning of the vaccination campaign, no such effect was apparent in the German sample, where an increase of 5% was found to require 3,250 EUR. Therefore, the observed effects of legal and monetary incentives are likely to be different for other populations and cultural backgrounds. Therefore, our results should be generalized with care. Moreover, the so-called compromise

effect could be a methodological issue complicating a straightforward interpretation of our results [30, 36, 37]. It has been shown that when presented a price list, participants are perceptually drawn to the center of the price list, making those options appear more attractive. This could bias the elicited amounts. However, since an incentive of 200 EUR in another hypothetical experiment not using a price list design proved to be ineffective at increasing vaccination intentions in the German population [14], it seems unlikely that this effect can explain our findings. Nevertheless, fictitious incentives and assessment of vaccination intentions may not offer a perfect representation of real-life vaccination decision-making. Although intention usually predicts behavior, as discussed by Sheeran [38] and supported by recent evidence in the context of COVID-19 by Campos-Mercade et al. [18], there surely is a gap between the two, especially as the social desirability of accepting rewards for vaccination may be low. Therefore, the observed effects of legal and monetary incentives on vaccination intentions may be considered conservative estimates. Research conducted about a year after the presented study (and the large-scale rollout of vaccines to all Germans) suggests that the vast majority of unvaccinated individuals is unwilling to get vaccinated because they consider vaccination unsafe and identify with not being vaccinated [39]. Thus, it is unlikely that smaller payments are able to convince this rather extreme group. However, large incentives could still motivate some individuals to get the first shot. The present findings are important for future pandemics and recurring immunization: incentives could be used to promote booster uptake of already vaccinated individuals [40] and speed-up vaccinations in future pandemics.

When large monetary incentives are needed to increase vaccination uptake, ethical and economic concerns arise. Incentives rob the act of vaccination of its moral significance [28], possibly generating expectations of receiving payment for other vaccinations as well. Furthermore, large payments could increase vaccine hesitancy, because they may be perceived as compensation for severe adverse effects. Scholars also fear that large payments could be especially coercive to economically disadvantaged groups [25]. However, we could not find a link between financial worries and willingness to get vaccinated for a monetary reward. But in case policymakers decide to announce and introduce payments, they should also be paid retrospectively to already vaccinated participants to prevent vaccination-ready individuals from deferring immunization and to mitigate anger and backlash due to perceived unfairness. From an economic perspective, monetary incentives are only feasible if the benefits of higher vaccination rates outweigh the payment costs. In the case of Germany, increasing vaccination uptake by a few percentage points would require several hundred billion euros, challenging the efficiency of the measure.

Fortunately, our results suggest an alternative pathway to increasing vaccination uptake. As high levels of confidence and collective responsibility and low levels of complacency and calculation were related to willingness to be vaccinated without payment, improving these aspects should increase vaccination uptake at much lower costs. Therefore, communication efforts should highlight the safety and efficacy of vaccines. As previous research has shown, providing information about the prosocial impact of vaccination is important too [4]. When people realize that their own shots also protect those who cannot be vaccinated, such as children and individuals with an immunodeficiency, vaccination intentions increase. Recent research on influenza vaccination further indicates that scheduling appointments for shots and sending messages reminding individuals about vaccination opportunities prior to primary care visits could boost vaccination rates at low costs [41]. In conclusion, incentives may work, but the cost–benefit ratio seems questionable. Only if educational efforts and nudges are insufficient to increase vaccination uptake, payments could add relevant percentage points, given thorough ethical embedment of the measure and sufficient monetary power.

## Supporting information

**S1 Table. Determinants of getting vaccinated.**
(DOCX)

**S2 Table. Determinants of getting vaccinated with and without monetary incentive.**
(DOCX)

**S3 Table. Determinants of minimum required monetary incentives.**
(DOCX)

## Acknowledgments

The study was conducted as part of Germany's COVID-19 Snapshot Monitoring (COSMO), a joint project of the University of Erfurt (Cornelia Betsch [PI], Lars Korn, Philipp Sprengholz, Philipp Schmid, Lisa Felgendreff, Sarah Eitze), the Robert Koch Institute (RKI; Lothar H. Wieler, Patrick Schmich), the Federal Centre for Health Education (BZgA; Heidrun Thaiss, Freia De Bock), the Leibniz Institute of Psychology (ZPID; Michael Bosnjak), the Science Media Center (SMC; Volker Stollorz), the Bernhard Nocht Institute for Tropical Medicine (BNITM; Michael Ramharter), and the Yale Institute for Global Health (Saad Omer).

## Author Contributions

**Conceptualization:** Philipp Sprengholz, Luca Henkel, Cornelia Betsch.

**Data curation:** Philipp Sprengholz, Luca Henkel.

**Formal analysis:** Philipp Sprengholz, Luca Henkel.

**Investigation:** Philipp Sprengholz, Luca Henkel.

**Methodology:** Philipp Sprengholz, Luca Henkel.

**Project administration:** Cornelia Betsch.

**Resources:** Cornelia Betsch.

**Visualization:** Philipp Sprengholz, Luca Henkel.

**Writing – original draft:** Philipp Sprengholz, Luca Henkel.

**Writing – review & editing:** Cornelia Betsch.

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
