## [Decision Letter · Decision Letter 0]

23 Mar 2022

PONE-D-21-34556Payments and freedoms: Effects of monetary and legal incentives on COVID-19 vaccination intentions in GermanyPLOS ONE

Dear Dr. Sprengholz,

Thank you for submitting your manuscript to PLOS ONE. After careful consideration, we feel that it has merit but does not fully meet PLOS ONE’s publication criteria as it currently stands. Therefore, we invite you to submit a revised version of the manuscript that addresses the points raised during the review process.

We look forward to receiving your revised manuscript.

Kind regards,

Andrew T. Marshall, Ph.D.

Academic Editor

PLOS ONE

Journal Requirements:

2. Please provide additional details regarding participant consent. In the Methods section, please ensure that you have specified (1) whether consent was informed and (2) what type you obtained (for instance, written or verbal). If your study included minors, state whether you obtained consent from parents or guardians. If the need for consent was waived by the ethics committee, please include this information.

This work was supported by German Research Foundation (BE3970/12-1), Federal Centre for Health Education, Robert Koch Institute, Leibniz Institute for Psychology, Klaus Tschira Foundation, and University of Erfurt (no award/grant numbers).

This work was supported by German Research Foundation (BE3970/12-1), Federal Centre for Health Education, Robert Koch Institute, Leibniz Institute for Psychology, Klaus Tschira Foundation, and University of Erfurt (no award/grant numbers).

Additional Editor Comments:

Thank you for your submission! Please address all reviewer comments, particularly those related to previous literature, the novelty of this study, and educational campaigns (as discussed by Reviewer #2).

Reviewers' comments:

Reviewer's Responses to Questions

**Comments to the Author**

1. Is the manuscript technically sound, and do the data support the conclusions?

Reviewer #1: Yes

Reviewer #2: Yes

2. Has the statistical analysis been performed appropriately and rigorously? 

Reviewer #1: Yes

Reviewer #2: Yes

3. Have the authors made all data underlying the findings in their manuscript fully available?

Reviewer #1: Yes

Reviewer #2: Yes

4. Is the manuscript presented in an intelligible fashion and written in standard English?

Reviewer #1: Yes

Reviewer #2: Yes

5. Review Comments to the Author

Reviewer #1: This study experimentally examines the influence of two factors – the presence/size of a financial incentive and whether vaccination would free individuals from testing requirements to engage in a range of social activities – on willingness to vaccinate in Germany. The experiment finds little evidence that the former matters; it finds some evidence that incentives can increase vaccination uptake, but only very large incentives.

The experimental design is clear, and the analysis is clearly and succinctly presented. My main question/concern centers on the timing of the survey itself – in April of 2021. At the very least, the paper could do more to contextualize where the vaccination campaign stood in Germany at the time this survey went into the field. For example, I was initially surprised that >60% of unvaccinated respondents said they would receive a vaccine – until I went back and saw that the survey was fielded in April vs., say, November. The Discussion might engage a bit more the question of what lessons we can still draw about vaccination intention now when many more people have had the vaccine and the holdouts are perhaps even more steadfast in their opposition.

I found the null results for the “freedom” treatment – vaccination can allow you to do a range of things without having to show proof of a negative test – particularly interesting. It also seems to dovetail with observational data from other countries. For example, I believe the vaccination campaign had slowed considerably in Italy when a negative test within 72 hours allowed you to obtain a Green Pass and engage in a wide range of activities… but many holdouts were converted and got vaccinated with the introduction of the Green Pass rafforzato and when this (which requires vaccination – not just a negative test) was made mandatory for a range of activities. The Discussion might engage such policy discussions a bit more directly.

In sum, I think the study is well-done. My main suggestions are that the authors could do more to contextualize the findings and highlight for readers how these results from a relatively early stage of the vaccination campaign can still inform contemporary debates over boosters and the like.

Reviewer #2: This paper studies the effects of incentives on willingness to take the COVID-19 vaccine in Germany. The study was conducted in Germany in April of 2021. The study varied two types of incentives: monetary and legal incentives. First, across participants, it compared willingness to get vaccinates when being vaccinated provided more freedoms to when it did not. Second, within-subjects, it elicited participants’ willingness to vaccinate themselves for 22 potential monetary payments. The study finds that willingness to get vaccinated is not affected by the freedoms that it could provide, but that incentives of over 3,000 Euro increase willingness to vaccinate.

Comments

Understanding the effectiveness of policies that aim to increase vaccination rates is important for policy makers worldwide. This study focuses on Germany and elicits willingness to vaccinate for a sample of 782 individuals, who are representative of the German population in age, gender, and region in which they live.

The authors have done extensive work on willingness to vaccinate (for COVID-19). What is the contribution that this paper makes? The authors’ previous work focused on smaller incentives, is the main contribution of the present work to focus on higher incentives? How does this work differ from the work published in PNAS focusing also on Germany by Kluver et al. (2021)? This work showed that small incentives increased intentions to vaccinate.

What motivated the design of such high incentives? Are there examples of companies or governments paying such large amounts of money for individuals to get vaccinated? Would there be political support for such measures?

Could the design section provide more detail to answer the following questions?

- What effect sizes was the study powered to detect?

- What percentage of subjects switched multiple times overall in the price list? How are they treated in the main analyses (e.g., Figure 1 and Table 1)? Were there monetary values (or a treatment) for which switching occurred more often?

- The data was collected in April of 2021, at that point what percentage of the population had access to vaccination? Was there already a concern that there would be significant hesitancy in Germany?

Table 1 divides participants into 3 groups, why not analyze each decision and cluster standard errors at the individual level? A lot of data is not considered in the analyses with this approach. The coefficients of a multinomial logit are also more difficult to interpret. If the reason is that several subjects switched multiple times, the analyses could be conducted including and excluding these subjects, for example.

I fail to understand the extrapolated cost calculation (Section 3.4). What are the underlying assumptions? Would the incentives be paid to everyone? What are the dynamic issues that may arise (people may prefer to wait in anticipation of an incentive)?

The paper’s abstract and conclusion mention educational interventions. However, the paper does not study educational interventions, in the sense that it does not test whether increasing perceptions of safety and understanding the prosocial impact of the vaccine lead to lower hesitance. Perhaps it’s better to remove references to a policy recommendation that has not been tested?

6. PLOS authors have the option to publish the peer review history of their article (what does this mean?). If published, this will include your full peer review and any attached files.

Reviewer #1: No

Reviewer #2: No

---

## [Decision Letter · Decision Letter 1]

11 May 2022

Payments and freedoms: Effects of monetary and legal incentives on COVID-19 vaccination intentions in Germany

PONE-D-21-34556R1

Dear Dr. Sprengholz,

We’re pleased to inform you that your manuscript has been judged scientifically suitable for publication and will be formally accepted for publication once it meets all outstanding technical requirements.

Kind regards,

Andrew T. Marshall, Ph.D.

Academic Editor

PLOS ONE

Reviewers' comments:

Reviewer's Responses to Questions

**Comments to the Author**

1. If the authors have adequately addressed your comments raised in a previous round of review and you feel that this manuscript is now acceptable for publication, you may indicate that here to bypass the “Comments to the Author” section, enter your conflict of interest statement in the “Confidential to Editor” section, and submit your "Accept" recommendation.

Reviewer #1: All comments have been addressed

Reviewer #2: All comments have been addressed

2. Is the manuscript technically sound, and do the data support the conclusions?

Reviewer #1: Yes

Reviewer #2: Yes

3. Has the statistical analysis been performed appropriately and rigorously? 

Reviewer #1: Yes

Reviewer #2: Yes

4. Have the authors made all data underlying the findings in their manuscript fully available?

Reviewer #1: Yes

Reviewer #2: Yes

5. Is the manuscript presented in an intelligible fashion and written in standard English?

Reviewer #1: Yes

Reviewer #2: Yes

6. Review Comments to the Author

Reviewer #1: Thank you for these careful replies and edits. I appreciate the additional discussion and context, and I am now pleased to support publication.

Reviewer #2: I have read the reply letter and the manuscript. The revision addressed my comments. I have no further comments for the authors.

7. PLOS authors have the option to publish the peer review history of their article (what does this mean?). If published, this will include your full peer review and any attached files.

Reviewer #1: No

Reviewer #2: No

---

## [Editor Report · Acceptance letter]

13 May 2022

PONE-D-21-34556R1 

Payments and freedoms: Effects of monetary and legal incentives on COVID-19 vaccination intentions in Germany 

Dear Dr. Sprengholz:

I'm pleased to inform you that your manuscript has been deemed suitable for publication in PLOS ONE. Congratulations! Your manuscript is now with our production department. 

Kind regards, 

on behalf of

Dr. Andrew T. Marshall 

Academic Editor

PLOS ONE